# Effect of Increasing the Number of Substitutions on Physical Performance during Periods of Congested Fixtures in Football

**DOI:** 10.3390/sports11020025

**Published:** 2023-01-19

**Authors:** Abraham García-Aliaga, Adrián Martín-Castellanos, Moisés Marquina Nieto, Diego Muriarte Solana, Ricardo Resta, Roberto López del Campo, Daniel Mon-López, Ignacio Refoyo

**Affiliations:** 1Facultad de Ciencias de la Actividad Física y del Deporte (INEF—Sports Department), Universidad Politécnica de Madrid, 28040 Madrid, Spain; 2Department of Competitions and Mediacoach, LaLiga, 28043 Madrid, Spain

**Keywords:** GPS, physical demands, soccer, rule modification, lockdown, substitution window

## Abstract

(I) This study aimed to evaluate the impact on physical demands induced by FIFA’s new rule implemented based on the number of substitutions caused by COVID-19. (II) Sixty-six matches were analysed in peak periods (microcycles of three matches in a week) in the competition period before and after the pandemic. The variables collected were organised by team (22 from LaLiga^TM^ SmartBank 2019–2020) for a total of 132 team records and 1077 player performance reports using a multi-camera tracking system and Mediacoach^®^ software. Physical performance variables were analysed in the first half, second half and whole match, thus determining the individual and collective performances of the team. (III) This study shows how, despite the increase in substitutions allowed with the new rule, physical performance increased in some variables in the congested periods (e.g., total distance run and distance run in the first and second halves). Additionally, the players’ physical performance involved in a substitution was greater than it was for players who completed the game. (IV) The new substitution rule helps to maintain and even improve physical performance. This measure could improve intensity levels in both individual and team performance. It could even safeguard the physical integrity of the players by reducing the risk of injury, as fewer players have to play the full match.

## 1. Introduction

In March 2020, football seasons were interrupted worldwide to prevent the spread of the new coronavirus disease (COVID-19). Leagues restarted their football seasons after several weeks of interruption, causing overlap in their yearly calendars. This overlap has led to more congested schedules (e.g., five matches over 14 days), eventually leading to the development of cumulative fatigue in the players [1] and potentially increasing the risk of injury [2]. To minimise match overload and potential physical problems, the Fédération internationale de football association (FIFA^®^) has authorised an increase of five substitutions for each team per match instead of the usual three [3].

Therefore, the analysis of game rules can be instrumental in making changes to sports rules to improve games by making sports safer, healthier, fairer and more entertaining [4]. Recently, it has been shown that elite football has singularly higher rule-induced physical demands than other elite team sports. The rules of the game itself, especially including the reduced number of substitutions allowed, are a major contributing factor to the high demands of football [5]. Therefore, changes to the substitution rules could mitigate the overall demands of football [6].

In recent years, different regulations and rule changes have been reviewed in football. Each sport has its specific rules that determine which actions, materials and/or elements are allowed and which are not [4]. These rules detail the number of players on each team, the number of substitutions, the dimensions of the field and all aspects concerning the game. Analysis of the rules can be crucial when making rule changes to improve a sport [4]. Updating the rules can make the sport safer (e.g., shin pads, helmets for goalkeepers, etc.), healthier (e.g., refreshment break), fairer (e.g., video-assisted referees) or more entertaining (i.e., advantageous for sponsors and fans) to compete with other sports, including e-sports, which are growing rapidly [7].

Consequently, a rule change such as to rules on substitutions can induce an increase in intensity. The higher total physical demands in football compared to other team sports have been demonstrated in high-intensity variables [8], and a 30% increase in the sprint distance and technical variables have been revealed [9] relative to other team sports. These differences between sports are related because football is a complex sport of interaction between players (collaboration–opposition) in which players randomly transition between maximum (or near-maximum), high-intensity and multidirectional effort and longer periods of low-intensity activity [10]. Players typically run between 9 and 14 km in total during a match, with high-intensity running accounting for approximately 10% of that distance [11]. The physical demands placed on top-level football players have been extensively documented over the last few years [12,13,14]. From all these studies, it has been concluded that, on average, a player travels about 11 km during a match. However, due to the intermittent nature of the game, the total distance travelled represents an insufficient parameter to understanding the overall physical demands and therefore, the distance travelled at high speeds seems to be a better indicator of performance. Therefore, this variable has been related to the level of competition [15,16]. Furthermore, physical demands are affected by the positional role of the players in the team’s game formation. Consequently, central midfielders run the longest distances during matches, while wide midfielders cover the longest distances at high intensities [15,16].

These intermittent actions can be divided into different intervals using speed ranges: distance covered at a high intensity (14 km/h–21 km/h); distance covered at a very high intensity (21.1 km/h–24 km/h); and distance covered at a sprint (>24 km/h). The distances covered at higher speeds are progressively shorter as the speed ranges increase [17]. Furthermore, it has been suggested that the range of maximum speeds and the distance covered at very high intensities are crucial. Therefore, variables such as the number of sprints and the maximum speed achieved by an outfield player will be very important variables in analysing match performance data [18]. In the LaLiga^TM^ scenario, an increase has been observed in high-intensity running since 2012 in most players’ positions. The activities performed by the players were classified into categories according to the following speed thresholds [13]: (I) standing (0–0.6 km-h^−1^); (II) walking (0.7–7.1 km-h^−1^); (III) jogging (7.2–14.3 km-h^−1^); (IV) running (14.4–19.7 km-h^−1^); (V) high-intensity running (19.8–25.1 km-h^−1^); and (VI) sprinting (>25.1 km-h^−1^). 

Accordingly, football coaches have a large amount of data but a limited number of resources to change the course of a match [19]. Therefore, efficient substitution management could contribute to determining the outcome of a match [19,20]. In general, football coaches seem to have similar substitution patterns [21]. However, there are a variety of reasons for making substitutions, such as modifying tactical behaviours [22], counteracting fatigue [23], restructuring the team after a player has been dismissed, replacing yellow carded players [24], replacing injured or underperforming players [25] as well as providing playing time to players with fewer minutes and preventing fatigue accumulation among other team members [26]. Substitutes are typically introduced at halftime or during the second half of the game [25,26], although the differences in physical performance have not been studied using time criteria [26,27]. 

Moreover, the physical performance of players who enter the field of play replacing a teammate has been studied by different authors [11,23,28]. In general, under normal game conditions, substitutes have shown better data in high-intensity trips than those players they replace and those who play the entire game [9,28,29,30]. The position within the team structure [11,23,29], the result of the match [28] and the time of the season [29] can also influence the performance of substitutes. In addition, it has been shown that substitute players have a lower workload and volume, considering both factors, during games in periods of congestion and during training sessions themselves in these periods in which the intensity of daily work decreases [31]. This shows that substitutions increase the possibility of maintaining and even improving the physical performances of the players.

Considering these facts, there is no in-depth research on the impact of the possible physical demands induced by this rule change in football produced by COVID-19 and whether player performance has been affected. Understanding the possible differences between the rules that may influence sport-specific fatigue and performance could be useful in updating the rules and improving the sport. Therefore, this study aimed to assess the impact of rule-induced physical demands in football, focusing on substitution rules (including changes due to COVID-19) and analysing the impact of substitutions on player performance, with the aim to maintain the rule in the future to attract the interest of the spectator with more intense efforts in the game.

## 2. Materials and Methods

### 2.1. Participants

This study was carried out using the data from 22 “LaLiga^TM^ SmartBank” teams (Spanish second division of professional soccer). A total of 66 matches were analysed, corresponding to the 8th, 9th and 10th and numbers 32nd, 33rd and 34th league rounds. Rounds number 8, 9 and 10 were played in a 3-match window during the same week before the COVID-19 isolation period and before the change in the number of substitutions rule. In addition, rounds 32d, 33 and 34 were the first three rounds after the COVID-19 isolation period with a 3-match window during a week that applied the new five substitutions rule. Therefore, a total of 1256 player performance reports was collected. Full match activity was recorded. In both situations, VAR was in place. It is difficult to establish stoppage times determined by the video referee reviews themselves, but in the comparison presented in this study, the sample selected is limited, adjusting to the reality of the context (a single case before the stoppage). To avoid possible errors as much as possible, the sample was cleaned with up to 7 different exclusion criteria, to use the data in the most legitimate way possible.

Additionally, the following exclusion criteria were applied: (I) goalkeepers were excluded since their physical demands are not comparable with the rest of the specific positions; (II) sent-off players; and (III) matches that presented a lack of data (three matches from the 10th round). To analyse the impact of substitutions on performance, a comparison between the maximum number of possible changes before (i.e., 3) and after (i.e., 4–5) the rule modification was established. Those teams that presented one (*n* = 1) or two (*n* = 2) substitutions prior to the regulation change (IV) or one (*n* = 0), two (*n* = 0) or three (*n* = 8) changes after the regulation change (V) were eliminated. Players who received a red card were excluded from the sample. The rest of the data of players belonging to these matches were kept due to the very specific selection of the sample (congested periods; e.g., before the confinement, only one was found) and the dispersion it would entail when making comparisons (e.g., minutes a team spends with 10 players). Once the exclusion criteria were applied, the final sample was a total of 1077 players’ performance reports (Figure 1).

### 2.2. Variables

Following the previous literature [32], the performance variables analysed were: total distance covered; distance covered at high-intensity running (14 km/h–21 km/h); distance covered at very high-intensity running (21.1 km/h–24 km/h); distance covered at sprint (>24 km/h); amount of effort in the speed ranges of 21 km/h–24 km/h and >24 km/h; and maximum speed (km/h). Moreover, all variables were analysed for the 1st half, 2nd half and whole match [33,34,35]. These speed thresholds were restricted by the system used in the Spanish league (LaLiga^TM^).

### 2.3. Methodology

The official LaLiga^TM^ System (Mediacoach^®^, Madrid, Spain) was used to measure the variables. Mediacoach^®^ obtain the data using two validated systems: the Global Positioning System (GPS) [33,36] and a reference camera system (i.e., the VICON motion capture system) [37].

Taking into account the distribution used in the literature, substitutions were classified into 15 min temporal windows [26,27]. Solely the changes made in the second half were considered for this study, as those made in the first half did not present a representative sample (*n* = 8). Therefore, the following groups were established: whole game players (full match); sub 46–60 (substitution between 46 and 60 min); sub 61–75 (substitution between 61 and 75 min); and sub ≥ 76 (substitution from minute 76 onwards). Furthermore, substitutions during the halftime rest were considered in the sub 46–60 window. To estimate physical performance, those players who played the whole match were analysed individually. However, when a substitution was carried out, a single-match player was considered as the sum of both players, i.e., the player who started the game and the one who replaced him. In consequence, performance variables were calculated by adding up both players’ data, except for the maximum speed, which was calculated by taking the higher value of both players. The data collection and cleaning process is illustrated in Figure 1.

### 2.4. Statistical Analysis

The performance variables were described using the mean and standard deviation. The normal distribution of the variables was checked using the Kolmogorov–Smirnov test and the homogeneity of variance was tested using Levene’s test. Moreover, those performance variables’ values higher than three standard deviations were excluded [38]. 

An independent *t*-test was carried out to compare the number of substitutions before and after the substitutions rule change. When statistically significant differences were found, the effect size was estimated using Cohen’s d index (*d*), establishing two cut-off points: medium effect (0.30) [39] and large effect (0.60) [40].

One-way ANOVA was carried out to analyse the comparison between the substitution window before and after the substitution rule change. The post hoc analysis to set the differences between groups was carried out using the Bonferroni test. The effect sizes were determined using partial eta squared (η_p_^2^) and their interpretation was based on the following criteria [41]: small effects (0.01–0.06); moderate effects (0.06–0.14); and large effects (≥0.14). 

The level of significance was set as *p* < 0.01, although an alpha of 0.05 was used for post hoc comparisons. The collected data were studied using the software Statistical Package for the Social Science (SPSS, version 25.0; IBM Corporation, Armonk, New York, NY, USA).

## 3. Results

Differences can be observed in that the number of substitutions made by the teams after the rule change was significantly affected (t_130_ = 21.32; *p* < 0.001; d = 2.26), being higher when more substitutions were allowed (previously 2,94 ± 0,30; currently 4.58 ± 0.70).

### 3.1. Comparisons with the Previous Substitution Rule (Three Substitutions Max.)

With the previous rule, the comparison between substitution window groups showed differences in the total distance run (F_(3,535)_ = 16.82; *p* < 0.001, η_p_^2^ = 0.086), total distance run in the first half (F_(3,535)_ = 6.18; *p* < 0.001, η_p_^2^ = 0.034), total distance run in the second half (F_(3,535)_ = 27.23; *p* < 0.001, η_p_^2^ = 0.132), total distance run at 14–21 km/h (F_(3,535)_ = 13.51; *p* < 0.001, η_p_^2^ = 0.070), total distance run between 14–21 km/h in the first half (F_(3,535)_ = 6.30; *p* < 0.001, η_p_^2^ = 0.034) and second half (F_(3,535)_ = 20.47; *p* < 0.001, η_p_^2^ = 0.103), distance run at very high intensity at 21–24 km/h (F_(3,535)_ = 15.91; *p* < 0.001, η_p_^2^ = 0.082) in the first (F_(3,535)_ = 6.82; *p* < 0.001, η_p_^2^ = 0.037) and second half (F_(3,535)_ = 18.12; *p* < 0.001, η_p_^2^ = 0.092) and distance run at the sprint threshold (>24 km/h) for the whole game (F_(3,535)_ = 10.05; *p* < 0.001, η_p_^2^ = 0.092) and during the second half (F_(3,535)_ = 16.50; *p* < 0.001, η_p_^2^ = 0.085), as no differences were observed in the first half in this group (*p* = 0.049).

In relation to the amount of effort, differences were also found in total effort between running at 21–24 km/h (F_(3,535)_ = 15.21; *p* < 0.001, η_p_^2^ = 0.079) in the first (F_(3,535)_ = 6.47; *p* < 0.001, η_p_^2^ = 0.035) and second half (F_(3,535)_ = 18.33; *p* < 0.001, η_p_^2^ = 0.093) and the total time spent at a sprint (>24 km/h) in the whole game (F_(3,535)_ = 13.56; *p* < 0.001, η_p_^2^ = 0.071) and in the second half (F_(3,535)_ = 21.05; *p* < 0.001, η_p_^2^ = 0.106). The first half did not present significant differences for the total amount of time spent at a sprint >24 km/h (*p* = 0.006). In the case of the maximum speed for a completed game or the first or second half, no differences were observed in any comparison (*p* > 0.001). These differences can be observed in Table 1.

In addition, the Bonferroni post hoc test showed differences between substitution window groups in the total distance and total distance in the first and second half, with lower values for the whole game than the sub 61–75 and ≥ 76 groups (*p* < 0.001). Additionally, in the second half, differences were found among whole game players and sub 46–60 players, with greater values for the sub 46–60 group (*p* < 0.01). In the same lane, in total distance run at 14–21 km/h, the whole game players presented worse values than the sub 61–75 and ≥ 76 groups (*p* < 0.001). For the first half, the differences were similar, with significant differences in favour of the sub ≥ 76 (*p* < 0.01) and sub 46–62 (*p* < 0.05) groups; meanwhile, for the second half, whole game players scored the lowest values in all comparisons (sub 46–60, *p* < 0.01; sub 61–75 and sub ≥ 76, *p* < 0.001).

For distance covered at very high intensity, the sub 61–75 and sub ≥ 76 groups registered better performance than whole game players (*p* < 0.001). In the first half, these differences were maintained between the same groups (*p* < 0.01), but for the second half, whole game players scored lower values than the rest of the groups (sub 46–60, *p* < 0.01; sub 61–75 and sub ≥ 76, *p* < 0.001). Distance run at a sprint showed greater values in the sub 61–75 (*p* < 0.001) and sub ≥ 76 (*p* < 0.01) groups than whole game players, and the same distance in the second half substitution groups were higher than whole game players (sub 46–60, *p* < 0.01; sub 61–75 and sub ≥ 76, *p* < 0.001).

Comparisons between the amount of effort between 21 and 24 km/h highlighted a higher value for the sub 61–75 and sub ≥ 76 (*p* < 0.001) groups than the whole game group in total; the same group differences were shown in the first half (*p* < 0.01), but in the second half, all substitution groups demonstrated a higher amount of effort than whole game players (sub 46–60, *p* < 0.01; sub 61–75 and sub ≥ 76, *p* < 0.001). The number of total sprinting >24 km/h observed was lower for the whole game than sub 61–75 and sub ≥ 76 (*p* < 0.001) players. In the second half, all substitution groups scored greater values than the whole game players (sub 46–60, *p* < 0.01; sub 61–75 and sub ≥ 76, *p* < 0.001).

### 3.2. Comparisons with the Current Substitution Rule (Five Substitutions Max.)

After rule modification, the comparison between substitution window groups showed differences in the total distance (F_(3,532)_ = 32.82; *p* < 0.001, η_p_^2^ = 0.156), total distance in the first half (F_(3,532)_ = 9.76; *p* < 0.001, η_p_^2^ = 0.052), total distance in the second half (F_(3,532)_ = 54.91; *p* < 0.001, η_p_^2^ = 0.236), total distance at 14–21 km/h (F_(3,532)_ = 30.46; *p* < 0.001, η_p_^2^ = 0.147), total distance between 14–21 km/h in the first half (F_(3,532)_ = 10.01; *p* < 0.001, η_p_^2^ = 0.053) and second half (F_(3,532)_ = 49.23; *p* < 0.001, η_p_^2^ = 0.214), distance at very high intensity 21–24 km/h (F_(3,532)_ = 25.83; *p* < 0.001, η_p_^2^ = 0.127) in the first (F_(3,532)_ = 10.44; *p* < 0.001, η_p_^2^ = 0.056) and second half (F_(3,532)_ = 28.13; *p* < 0.001, η_p_^2^ = 0.137) and distance at the sprint threshold (>24 km/h) for the whole game (F_(3,532)_ = 12.01; *p* < 0.001, η_p_^2^ = 0.063) and the second half (F_(3,532)_ = 15.41; *p* < 0.001, η_p_^2^ = 0.080), since no differences were observed in the first half (*p* = 0.002) for this group.

In relation to the amount of effort, differences were also found in total efforts between 21–24 km/h (F_(3,532)_ = 25.60; *p* < 0.001, η_p_^2^ = 0.126) in the first (F_(3,532)_ = 10.24; *p* < 0.001, η_p_^2^ = 0.055) and second half (F_(3,532)_ = 30.59; *p* < 0.001, η_p_^2^ = 0.147) and the total time spent at a sprint (>24 km/h) in the whole game (F_(3,532)_ = 15.81; *p* < 0.001, η_p_^2^ = 0.082), first half (F_(3,532)_ = 7.18; *p* < 0.001, η_p_^2^ = 0.039) and second half (F_(3,532)_ = 19.04; *p* < 0.001, η_p_^2^ = 0.097). No differences were found in the maximum speed for the whole game or the first or second half (*p* > 0.001). These differences are shown in Table 2.

Group comparisons showed differences in the total distance. Players who completed the whole game achieved lower values than the rest of the groups in total distance and total distance in the first and second halves (*p* < 0.001), except for the total distance in the first half between the whole game and sub 46–60 groups (*p* < 0.01). In the case of the total distance run between 14 and 21 km/h, in the first and second halves, differences were observed with lower values for the whole game players than the rest of the groups, all with a *p* < 0.001, with one exception: total distance between 14 and 21 km/h in the first half among the whole game and sub 46–60 groups (*p* < 0.05).

Moreover, the comparison of distance run at very high intensity showed that whole game players had lower values than the rest of the groups (*p* < 0.001), whereas the sub 61–75 group presented greater values than the sub ≥ 76 group (*p* < 0.01). In the first half, sub 61–75 players scored numbers higher than the rest of the groups (whole game, *p* < 0.001; sub 46–60 and sub ≥ 76, *p* < 0.05), and in the second half, the performance of whole game players was worse than the other groups (*p* < 0.001) and sub 61–75 players presented greater values than sub ≥ 76 ones (*p* < 0.05). For distance covered at the sprint threshold, differences were found between the whole game and substitution groups in the comparisons with the whole game (sub 46–60, *p* < 0.05; sub 61–75, *p* < 0.001; and sub ≥ 76, *p* < 0.01) and second half (sub 46–60, *p* < 0.01; sub 61–75 and sub ≥ 76, *p* < 0.001) groups.

Finally, total effort between 21 and 24 km/h presented differences between the whole game and substitutions groups (*p* < 0.001), being greater for substituted players, and sub 61–75 scored higher values than sub ≥ 76 (*p* < 0.01). In the first half, sub 61–75 presented the highest values (whole game, *p* < 0.001; sub 46–60, *p* < 0.01; and sub ≥ 76, *p* < 0.05); meanwhile, in the second half, whole game players’ values were lower than the rest of the groups (*p* < 0.001 in all comparisons). In addition, the sub 61–75 group obtained greater values than the sub ≥ 76 group (*p* < 0.05) in the second half. For the number of sprints (>24 km/h), the sub 61–75 and sub ≥ 76 groups scored higher values than the whole game group (*p* < 0.001 and *p* < 0.01, respectively). In addition, sub 61–75 presented greater values than sub 46–60 and sub ≥ 76 (*p* < 0.05). In the halftime analysis, the first half showed that sub 61–75 players achieved higher values than whole game (*p* < 0.001) and sub 46–60 (*p* < 0.05) players, whereas in the second half, whole game players presented lower values than the rest of the groups (sub 46–60, *p* < 0.01; sub 61–75 and sub ≥ 76, *p* < 0.001).

## 4. Discussion

This study aimed to evaluate the impact on physical intensity variables before and after the COVID-19 isolation period of the new five substitutions rule in football, considering, in particular, the substitution window. The main finding of this study is that despite an increase in the number of substitutions allowed with the new rule, physical performance presented increases in a few variables during congested periods. Only the total distance run in the first and second half and distance run between 14 and 21 km/h in the second half were higher after the isolation period. In addition, the physical performance scores for players involved in a substitution were greater than for players who completed the whole game.

According to the results, it appears that the increase in substitutions in football to five players due to COVID-19 increases the percentage of player substitutions (~21.7%), but this number is still small considering the high physical demands. In addition, the substituted player cannot return to the same match in football, whereas they can in other team sports. The substitutes who participated in the matches (i.e., non-starters) had a lower workload (internal and external), taking into account both matches and training (internal and external), during six matches in 21 days (i.e., congested schedules) [31]. Their data suggest that matches are crucial in the training process, which means that substitutes may be detrained.

It is for this reason that the management of the number of substitutions allowed in a sport is a factor that can be decisive. In the case of football, substitutions are reduced and this encourages the high demands demonstrated in this sport [5]. Compared to other team sports, it has been shown that football has higher total physical demands [9]. All actions in the sport of football are of maximum intensity; they are shorter than most sports but are becoming more and more demanding. Football is a sport of intermittency where actions occur at maximum speed. Players take advantage of stoppages in play to recover and return to maximum intensity efforts. Therefore, if the number of substitutions is increased, there are more options to incorporate players into the game who can maintain their individual physical performance as well as the performance of the collective team. In addition, the sport’s evolution and its increase in playing speed have been demonstrated by increases of approximately 30% in sprint distance [9]. This increase in physical demands has been associated with greater cumulative fatigue and higher injury rates [42,43]. This is combined with the congestion of competition calendars, where teams have to play matches within 72–96 h of each other [2,44]. Furthermore, it has been shown that substitutes who participated in matches presented a lower workload in congested matches, so matches are crucial in the training process, meaning that substitutes may be detrained [31]. Therefore, the new regulations, conditioned by COVID-19, would theoretically help to reduce the risk of injury, especially considering that the sum of physical performance records is greater when players are replaced more often, as shown in the results of this research. Given this appreciation, higher values were expected in the study for variables that could be influenced by a short training period post-COVID-19, the accumulation of matches or psychological factors after isolation [45]. 

Following the idea of increasing the substitutions, some authors have even called for unlimited substitutions, as in other sports [5], arguing that this measure could mitigate a drop in the intensity of matches, especially in the second half [1,30].

In this study, the results seem to be in line with other authors who also showed an improvement in performance with increased substitutions, where the substitutes covered a greater distance of high-intensity running [9]. Moreover, changes during the match can provide a physical and/or tactical boost [28]. Substitutions in soccer matches usually take place at halftime or during the second half, except in cases of injury. The purpose of these changes is to reduce the fatigue of the starting players and to modify the tactical framework according to the situation of the match [25,30]. To achieve this objective, it is considered that the substitutes must provide a physical performance equal or even superior to their starting teammates [28,30]. In this practice, previous authors found that substitutes covered a greater total distance and at a higher speed than the players who played the full match or the players they replaced [16,23]. Furthermore, it has been shown that both early and late substitutes exerted more effort in sprints and spent more time and covered more distance at medium and high speeds per minute of play time than the full-match players [46]. 

These data are related to the findings of this study, where the players who completed the entire match presented worse data than those who were substituted in during the game. In addition, the analysis with the previous rules showed differences in the performance peak in sub ≥ 76 group for total distance run and the total distance run between 14 and 21 km/h. In both halves, the distance run at very high intensity, during sprints and the amount of effort in these ranges highlighted a better performance for players replaced between 61 and 75 min. However, with the new rule modification, it has been observed that the sub 61–75 group presented the greatest performance values of all groups, except in the second half between 14 and 21 km/h, where the sub 46–60 group showed better numbers. The fact that the best physical performance scores were associated with the substitution groups in the final stretches of the match could be explained by the players’ need for time to adapt themselves to the match’s rhythm or to integrate into the collective tactical behaviours [47].

Therefore, for future research, it is necessary to consider the positions of the players and the specific moments and situations during the match in which the substitution is made, because substitutions can be defensive if the team is winning, offensive if the team is looking for victory or neutral if the changes are made player-by-player in the same position without influence on the tactical formation. Previous studies related to physical performance have also shown that substitute players showed a greater total distance covered and number of sprints by playing time in all playing positions than players who were substituted or who completed the entire match [48].

As can be seen from our results, there were not always improvements in the second half of the match. These data may be due, firstly, to the COVID-19 situation itself, during which the training methods were changed and led to a reduction in intensity in general terms compared to the second half of the competition [32,49,50]. On the other hand, the improvement in some of the data found in the first set of rounds concerning the second rounds could be related to the fact that the analysis of match days 32 to 34 comprised the first matches after league matches were paused, so the rhythm of the competition was not yet fully established [51,52]. Even the improvements in maximum speed could indicate that by knowing that there are a greater number of changes, players can increase their effort intensity when they are on the field since a teammate can replace them in the second half and match or increase their intensity. Another fact that could motivate the improvement in the pre-isolation data is the boost that playing at home could have given to the home teams since, after the pandemic, the attendance of the public was forbidden.

However, some limitations have been found in this study, such as the days chosen. It was necessary to choose a week with a higher density of matches, with the intent that it would be the most demanding; however, in the case of LaLiga^TM^ SmartBank, before the lockdown, there was only one week with these characteristics. In addition, the fact that the data are from LaLiga^TM^ SmartBank teams means that these teams are not used to playing matches during the week, as they do not usually play European competitions or take breaks for international commitments with their national teams, so they may not be familiar with these periods of match congestion. In addition, it would also be interesting to perform the same study in a non-congested period to see if changes are apparent between the two sets of matches and to make comparisons for 15 min periods. Of course, with a higher sample size, it would be possible to draw more conclusions. Finally, it would be very interesting to see what would happen in each playing position.

With the above, coaches must have the ability to deal with the mini matches that occur in the game and address their substitution tendencies according to the intensities that occur in a match, in order to increase the pace of play and the tactical load in their approaches.

## 5. Conclusions

The changes in in the number of substitutions in the LaLiga^TM^ SmartBank after the COVID-19 outbreak significantly affected performance, perhaps more moderately than might have been expected. This was probably due to good planning and monitoring by the coaches during the confinement period and upon returning to training. The physical data obtained generally improved in those players substituted in the final part of the match (>60 min) compared to their teammates who played the whole match. The data relating to distances covered were better after the period of confinement. Therefore, the new regulation increasing the number of substitutions helped teams to maintain and even improve their physical performance, even after a period when training conditions were not the usual for these football teams. Therefore, this measure could improve the intensity levels of their teams in a normal season, even improving the health of the athletes in terms of reduced risk of injury or even reduced mental fatigue. All indications suggest that this measure will continue after the pandemic, allowing coaches to make better use of the resources they have in their teams to obtain better performance during official competitions that have a high density of top-level players. Being able to use a larger number of players in a match is therefore an advantage. More changes means more fit players, better match rhythm, less injuries, more frequent rotation of players and more possibilities for coaches. 

## Figures and Tables

**Figure 1 sports-11-00025-f001:**
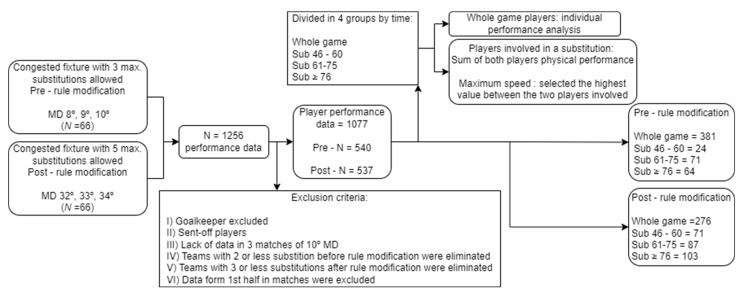
Data collection and cleaning carried out in this research.

**Table 1 sports-11-00025-t001:** Comparison between variables by substitution groups of players with the previous rule.

		Whole Game (*n* = 381)		Sub 46–60 (*n* = 24)		Sub 61–75 (*n* = 71)		Sub ≥ 76 (*n* = 64)				
		M	SD	M	SD	M	SD	M	SD	p	η_p_^2^	I
Total distance (m)		10,030.43	810.15	10,405.25	711.32	10,548.88 ^A^***	670.05	10,595.72 ^A^***	698.79	<0.001	0.086	M
	1st half (m)	5034.56	435.13	5077.16	411.49	5200.23 ^A^***	360.72	5232.85 ^A^***	407.85	<0.001	0.034	S
	2nd half (m)	4995.87	423.55	5328.09 ^A^**	456.74	5348.65 ^A^***	397.54	5362.87 ^A^***	378.35	<0.001	0.132	M
Total distance 14–21 km/h (m)		2096.85	527.66	2317.65	449.70	2392.87 ^A^***	447.56	2437.90 ^A^***	485.29	<0.001	0.070	M
	1st half/min (m)	1082.77	294.46	1114.10	253.24	1188.77 ^A^*	255.13	1224.19 ^A^**	276.14	<0.001	0.034	S
	2nd half/min (m)	1014.08	266.61	1203.55 ^A^**	251.02	1204.10 ^A^***	242.53	1213.71 ^A^***	258.41	<0.001	0.103	M
Distance Very High Intensity 21–24 km/h (m)		261.95	104.48	303.49	87.96	335.35 ^A^***	93.11	329.84 ^A^***	113.86	<0.001	0.082	M
	1st half 21–24 km/h (m)	131.52	60.47	137.10	50.33	159.64 ^A^**	63.58	159.27 ^A^**	70.08	<0.001	0.037	S
	2nd half 21–24 km/h (m)	130.43	60.66	166.39 ^A^**	59.42	175.71 ^A^***	53.87	170.57 ^A^***	65.09	<0.001	0.092	M
Total efforts 21–24 km/h		22.86	8.60	27.08	7.23	28.51 ^A^***	7.70	28.36 ^A^***	8.94	<0.001	0.079	M
	1st half 21–24 km/h	11.58	4.97	12.25	4.36	13.82 ^A^**	5.04	13.80 ^A^**	5.65	<0.001	0.035	S
	2nd half 21–24 km/h	11.27	4.90	14.83 ^A^**	5.01	14.69 ^A^***	4.35	14.56 ^A^***	4.94	<0.001	0.093	M
Distance Sprint >24 km/h (m)		238.96	136.99	286.36	149.08	322.22 ^A^***	155.27	307.66 ^A^**	150.30	<0.001	0.053	S
	1st half >24 km/h (m)	120.47	78.55	122.51	70.95	141.22	90.87	145.33	89.29	0.049		
	2nd half >24 km/h (m)	118.49	77.47	163.85 ^A^*	103.07	181.00 ^A^***	86.53	162.33 ^A^***	82.18	<0.001	0.085	M
Total number of sprints > 24 km/h		13.19	6.76	16.58	6.74	17.86 ^A^***	7.72	17.20 ^A^***	8.01	<0.001	0.071	M
	1st half >24 km/h	6.76	3.88	7.13	3.55	8.09	4.46	8.30	4.60	0.006		
	2nd half >24 km/h	6.44	3.80	9.46 ^A^**	4.53	9.78 ^A^***	4.36	8.91 ^A^***	4.30	<0.001	0.106	M
Max. Speed (km/h)		30.47	1.79	30.73	1.54	30.77	1.65	30.73	1.78	0.422		
	1st half (km/h)	29.61	2.05	29.81	1.82	29.65	2.06	29.75	2.13	0.925		
	2nd half (km/h)	29.59	2.05	29.81	1.90	30.17	1.85	29.96	1.80	0.098		

Significant differences are illustrated as: A = significant differences with “whole game” players, *I* = effect size interpretation. S = small effect size; M = moderate; L = large. Post hoc comparisons are highlighted: * *p* < 0.05; ** *p* < 0.01; *** *p* < 0.001.

**Table 2 sports-11-00025-t002:** Comparison between variables by substitution groups of players with the current rules.

		Whole Game (*n* = 276)		Sub 46–60 (*n* = 71)		Sub 61–75 (*n* = 87)		Sub ≥ 76 (*n* = 103)				
		*M*	*SD*	*M*	*SD*	*M*	*SD*	*M*	*SD*	*p*	η_p_^2^	*I*
Total distance (m)		10,105.24	862.02	10,760.46 ^A^***	761.67	10,847.60 ^A^***	635.72	10,756.56 ^A^***	771.49	<0.001	0.156	L
	1st half (m)	5053.10	465.69	5254.50 ^A^**	473.23	5280.85 ^A^***	393.86	5264.89 ^A^***	482.46	<0.001	0.052	S
	2nd half (m)	5052.14	455.87	5505.97 ^A^***	443.87	5566.75 ^A^***	354.36	5491.67 ^A^***	374.35	<0.001	0.236	L
Total distance 14–21 km/h (m)		2008.20	515.82	2411.89 ^A^***	461.28	2430.92 ^A^***	392.34	2399.76 ^A^***	517.06	<0.001	0.147	L
	1st half/min (m)	1025.61	279.42	1137.05 ^A^*	274.04	1167.15 ^A^***	249.20	1163.43 ^A^***	312.61	<0.001	0.053	S
	2nd half/min (m)	982.58	270.88	1274.84 ^A^***	303.30	1263.77 ^A^***	209.40	1236.33 ^A^***	242.10	<0.001	0.217	L
Distance Very High Intensity 21–24 km/h (m)		251.17	99.18	314.43 ^A^***	106.57	352.09 ^A^***^. D^**	104.43	299.07 ^A^***	98.18	<0.001	0.127	M
	1st half 21–24 km/h (m)	126.97 ^C^***	58.71	140.28 ^C^*	66.72	167.78	63.63	140.85 ^C^*	53.85	<0.001	0.056	S
	2nd half 21–24 km/h (m)	124.20	57.82	174.15 ^A^***	74.45	184.31 ^A^***^. D^*	66.69	158.22 ^A^***	61.80	<0.001	0.137	M
Total efforts 21–24 km/h		22.15	8.31	27.34 ^A^***	8.74	30.49 ^A^***^. D^**	8.56	26.08 ^A^***	8.04	<0.001	0.126	M
	1st half 21–24 km/h	11.40 ^C^***	5.01	12.21 ^C^**	5.33	14.74	4.90	12.48 ^C^*	4.30	<0.001	0.055	S
	2nd half 21–24 km/h	10.75	4.67	15.13 ^A^***	6.11	15.76 ^A^***^. D^*	5.28	13.60 ^A^***	5.18	<0.001	0.147	L
Distance Sprint >24 km/h (m)		226.60	140.60	279.12 ^A^*	150.36	325.34 ^A^***	154.15	283.47 ^A^**	146.43	<0.001	0.063	M
	1st half >24 km/h (m)	117.92	81.56	130.75	81.62	157.01	96.36	137.00	87.05	0.002		
	2nd half >24 km/h (m)	108.68	76.59	148.37 ^A^**	99.12	168.32 ^A^***	82.40	146.47 ^A^***	78.13	<0.001	0.080	M
Total number of sprints > 24 km/h		12.64 ^C^***	7.12	14.99 ^C^*	7.02	18.38 ^A^***	7.54	15.61 ^A^**^. C^*	6.83	<0.001	0.082	M
	1st half >24 km/h	6.60	4.19	6.97	3.95	8.98 ^A^***^. B^*	4.91	7.57	4.03	<0.001	0.039	S
	2nd half >24 km/h	6.03	3.84	8.01 ^A^**	4.69	9.40 ^A^***	4.01	8.04 ^A^***	3.91	<0.001	0.097	M
Max. Speed (km/h)		30.41	1.79	30.75	1.83	30.75	1.58	30.84	1.92	0.110		
	1st half (km/h)	29.72	2.15	29.54	1.77	29.68	2.09	29.87	2.09	0.787		
	2nd half (km/h)	29.26	2.11	29.85	2.29	30.08	1.71	29.99	2.17	0.001		

Significant differences are illustrated as: A = significant differences with “whole game” players, B = significant differences with “substitution 46–60” players, C = significant differences with “substitution 61–75” players and D = significant differences with “substitution ≥ 76” players. *I* = effect size interpretation. S = small effect size; M = moderate; L = large. Post hoc comparisons are highlighted: * *p* < 0.05; ** *p* < 0.01; *** *p* < 0.001.

## Data Availability

Not applicable.

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
