# Peer review of "Effect of Increasing the Number of Substitutions on Physical Performance during Periods of Congested Fixtures in Football"

_sports, 2023, doi:10.3390/sports11020025_

Round 1

Reviewer 1 Report

Introduction

The introduction lacks internal logic for the subjects being introduced. After the second paragraph, which ended by showing the importance of considering the game rules when analyzing players’ performance, a lot of match-related data was shown without any mention to rule changes. Also, some paragraphs are really generic (for example, see paragraph 3, in which a “30% increase is mentioned; but increase compared to what?)

L53-54: This is also generic. Running nine is entirely different from running 14km in a match. Also, age-related, position-related, and level-related differences must be considered.

Do substitutions increase or decrease physical performance according to previous studies? If there is a consensus, it could be argued that the hypothesis at the end of the introduction is plausible (increasing substitutions increase/decrease physical performance).

Methods

L114-115: it is unclear whether the match period was considered an independent variable. If so, further statistical analysis is required.

Congested fixtures were adopted. However, how do the authors ensure the same players were used in these matches? Because if there are different players, this is not a congested fixture at all. Instead, all the matches pre and post COVID could be selected as sample for this research.

L121-131: this increases the doubts elicited in the previous point. It is not clear the independent variable investigated. Here, instead of considering match periods, their 15-minute intervals were introduced. Also, if the aim is to compare matches pre and post-rule change, it is not clear why splitting the data is required. This reduces the sample size within each group and reduces the power of the statistical analysis.

Based on the abovementioned commentary, the current methodology is not reproducible and requires further description.

Regarding the statistical analysis, as there are multiple dependent variables, a MANOVA is much more recommended than an ANOVA to control the type I error.

Also, instead of considering all the dependent variables in one statistical model, the authors run separate tests for each independent variable, which also increases the probability of error.

Reference 37 is used to justify effect size thresholds. However, this is not a statistical article, therefore, it has not proposed thresholds. The classical Cohen’s d thresholds are different from those presented in the current paper, and there is no reason for it.

Results

Some results are hard to understand. For example, in table 2: how was this data organized? If there were multiple substitutions within a period, how were they taken into account? And what happened in case there were no substitutions at all in a period (which is quite common for the 45-60 interval)? Much more explanation on this topic’s methodology is required to allow the reader to really understand what is being compared here.

Discussion

The discussion, as in the introduction, lacks a logical order for the topics. Also, links between the paragraphs should be included.

L265: the impact “on” suits better

L271-272: which table does present the data related to the comparison between substitute and regular players?

L274-275: this makes no sense to me. How do decreasing substitutions encourage high demands? The opposite seems to make more sense, as players tend to reduce the efforts to support the whole game as not every player can be replaced. 

Author Response

Dear Reviewer

First of all, we would like to thank you for your decision on the manuscript and for all the considerations and comments made to improve the quality of our article. The reviewer's comments are marked in black, the authors' response in blue and the changes made to the paper in red.

Introduction

The introduction lacks internal logic for the subjects being introduced. After the second paragraph, which ended by showing the importance of considering the game rules when analyzing players’ performance, a lot of match-related data was shown without any mention to rule changes. Also, some paragraphs are really generic (for example, see paragraph 3, in which a “30% increase is mentioned; but increase compared to what?)

The introduction has tried to set a general background to the situation because there are no studies that have analysed the relevance of the number of substitutions in terms of actuality. Therefore, an attempt has been made to create a context from what little there was. In response to your concern, aspects of the introduction have been changed and more information has been added. In addition, English has been revised throughout the manuscript and ideas and paragraphs have been merged. 

Changes made from line 47 onwards

Therefore, in recent years, different regulations and rule changes have been reviewed in football. Each sport has its specific rules, which determine which actions, materials and/or elements are allowed and which are not [7]. This details the number of players in each team, the number of substitutions, the dimensions of the field and all aspects concerning the game. Analysis of the rules can be crucial when making rule changes to improve a sport [7]. Updating the rules can make the sport safer (e.g. shin pads, helmets for goalkeepers...), healthier (e.g. refreshment break), fairer (e.g. video assistant referee) or more entertaining (i.e. advantageous for sponsors and fans) to compete with other sports, including e-sports, which are growing rapidly [8].

Consequently, a rule change such as substitutions can induce an increase in intensity. The higher total physical demands in football have been demonstrated in high-intensity variables, compared to other team sports [9], and revealed a 30% increase in the sprint distance and technical variables [10] relative to other team sports. These differences between sports are related because football 

L53-54: This is also generic. Running nine is entirely different from running 14km in a match. Also, age-related, position-related, and level-related differences must be considered.

We fully agree with your consideration. An attempt has been made to explain it in a better way in line 64:

The physical demands placed on top-level football players have been extensively documented over the last few years ([13–15]. From all these studies it has been concluded that, on average, a player travels about 11 km during a match. However, due to the intermittent nature of the game, the total distance travelled represents an insufficient parameter to understand the overall physical demands and therefore the distance travelled at high speeds seems to be a better indicator of performance. Therefore, this variable has been related to the level of competition [16,17]. Furthermore, physical demands are affected by the positional role of the players in the team's game formation. Consequently, central midfielders run the longest distances during matches, while wide midfielders cover the longest distances at high intensities [16,17].

Do substitutions increase or decrease physical performance according to previous studies? If there is a consensus, it could be argued that the hypothesis at the end of the introduction is plausible (increasing substitutions increase/decrease physical performance).

The changes do indeed increase the likelihood of maintaining and even increasing the physical performance of players, and previous studies have corroborated this. This has been further explained and detailed.

Moreover, the physical performance of players who enter the field of play replacing a teammate has been studied by different authors [24,29–31]. In general, and under normal game conditions, substitutes have shown better data in high-intensity trips than those players they replace and those who play the entire game [30–33]. The position within the team structure [24,29–31] the result of the match [30] and the time of the season [31] can also influence the performance of substitutes. In addition, it has been shown that substitute players have a lower workload and volume, considering both factors, the games in periods of congestion and the training sessions themselves in these periods in which the intensity of daily work decreases [34]. This shows that substitutions increase the possibility of maintaining and even improving the physical performances of the players.

Methods

L114-115: it is unclear whether the match period was considered an independent variable. If so, further statistical analysis is required.

Congested fixtures were adopted. However, how do the authors ensure the same players were used in these matches? Because if there are different players, this is not a congested fixture at all. Instead, all the matches pre and post COVID could be selected as sample for this research.

L121-131: this increases the doubts elicited in the previous point. It is not clear the independent variable investigated. Here, instead of considering match periods, their 15-minute intervals were introduced. Also, if the aim is to compare matches pre and post-rule change, it is not clear why splitting the data is required. This reduces the sample size within each group and reduces the power of the statistical analysis.

Based on the abovementioned commentary, the current methodology is not reproducible and requires further description.

Regarding the statistical analysis, as there are multiple dependent variables, a MANOVA is much more recommended than an ANOVA to control the type I error.

Also, instead of considering all the dependent variables in one statistical model, the authors run separate tests for each independent variable, which also increases the probability of error.

Reference 37 is used to justify effect size thresholds. However, this is not a statistical article, therefore, it has not proposed thresholds. The classical Cohen’s d thresholds are different from those presented in the current paper, and there is no reason for it.

Regarding the method and the use of different time periods, we believe it is logical to think that, with most of the changes being made in the second half, we believe that taking into account the demands of the first half, the whole match and the second half (the period of performance of the changes) could facilitate the interpretation of the study, as well as give rise to others who can analyse the physical demands of the first half to explain the number of changes and the timing of the changes.

The 15-minute distribution has previously been used to structure periods equally (e.g. in notational performance analysis). In this case it is applied because we consider that it is not the same situation for a player who can play between 15 and 1 minutes, substituted at the end, as for a player who enters at half-time, and that performance could be affected by this (so groups were established). Another perspective that could be used would be the averaged values per minutes played, which are based on a similar approach, although the groups should be maintained by categorisation.

In relation to the question about congested periods, the number of players that a team can field is limited, it is not possible to recapitulate a large sample of each team because normally the maximum number of licences is between 24 - 30. Furthermore, the use of statistics with a related sample would presuppose the use of exactly the same players and the same changes, which is rarely the case, even though there may be a "block" of core players, who make up the structure of the team.

We have performed a MANOVA analysis following your indications, the problem we found is that the assumption of multicollinearity is not met, so performing the analysis in this way could involve errors as well. Nevertheless, we performed a pre-analysis and observed that the differences were maintained at the general level, with the data for the group of players who completed the match being lower overall. We opted to reduce the p level in the overall statistic (< .001), keeping p < .05 for the post hoc comparisons, and thus reduce the probability of error.

Reference number 37 was modified as it was in relation to the previous line.

Results

Some results are hard to understand. For example, in table 2: how was this data organized? If there were multiple substitutions within a period, how were they taken into account? And what happened in case there were no substitutions at all in a period (which is quite common for the 45-60 interval)? Much more explanation on this topic’s methodology is required to allow the reader to really understand what is being compared here.

In relation to the comments in the results section, an explanatory graphic has been added to explain how the data collection, data cleaning and analysis was carried out to facilitate replicability and understanding of our research. We believe that the inclusion of this figure and the restructuring of the explanation of the process can facilitate the problems discussed above. As for the tables, we have eliminated one comparison because we considered that it made the explanation of the results more complex. At present, only the one before and after the changes in the regulations are available. We believe that they add clarity to the study, and other reviewers have commented that the information can be appreciated correctly and that they are tables that can be understood. If there are any modifications that need to be made, please indicate them in order to clarify the information that is more complex.

We agree that it is a long section to interpret with respect to the text, but following the traditional writing methodology for this section, we must report all the results in writing, although we believe that the tables faithfully represent the summary of this section.

Thank you for your consideration

Discussion

The discussion, as in the introduction, lacks a logical order for the topics. Also, links between the paragraphs should be included.

The English has been modified and the links between ideas and paragraphs have been corrected. Information has been added from the line 299:

The increase in substitutions in football due to COVID-19 (five players) increases the percentage of player substitutions (~21.7%), but is still small considering the high physical demands. In addition, the substituted player cannot return to the same match in football, whereas in other team sports they can. The substitutes who participated in the matches (non-starters) had a lower load (internal and external), taking into account both matches and training (internal and external), taking into account both matches and training, during six matches in 21 days (i.e. congested schedules) [34]. Their data suggest that matches are crucial in the training process, which means that substitutes may be detrained.

L265: the impact “on” suits better

Done

L271-272: which table does present the data related to the comparison between substitute and regular players?

We have removed the previous table 1. We have left only two tables: both present a comparison between whole game players and substitution changes based on minutes of substitution. The current first table presents the differences with the previous regulation (maximum 3 changes) and the second table presents the comparison with the current regulation (maximum 5 changes).

L274-275: this makes no sense to me. How do decreasing substitutions encourage high demands? The opposite seems to make more sense, as players tend to reduce the efforts to support the whole game as not every player can be replaced. 

In response to your suggestion, reducing the number of substitutions does place greater physical demands on the players. It is not possible for players to self-regulate and if there were fewer substitutions they would maintain their strength throughout the match. Information has been added to line 310 for clarity.

The sport of football is evolving towards the opposite, all actions are of maximum intensity, and they are shorter and shorter but more and more demanding. Football is a sport of intermittency where actions are at maximum speed. Players take advantage of stoppages in play to recover and return to maximum-intensity efforts. Therefore, if the number of substitutions is increased, there are more options to incorporate players into the game who maintain that individual physical performance but also the collective one.

Reviewer 2 Report

Firstly, I'd like to say that I really like the research question that you have approached in the article. I think it is a really important question that has direct use within the field. 

I think there are some changes that need to be made to improve the quality of the article and to effectively reflect the quality of the research undertaken:

Abstract: 

I feel the abstract needs to be re-written. This section doesn't summaries the article well enough and is generally poor. I think you need to remove the mention of the 'spectacle'. This is a scientific article. This should be focused on changes in quantifiable measures like the physical or technical outputs of players, not a subjective assessment. The spectacle of the game could be very different simply because of the team which a person supports. 

Introduction: 

In general, this section doesn't read well despite some good parts to it. The english needs significant revision to make this section read better. 

I'm not sure about the velocity thresholds that you have used for high intensity distance, very high intensity, and sprint distance. This is different to those universally used in soccer and utilised in most research in these areas. 

Lines 75-78 - I think you need to look in to this comment further. If you are looking at the difference between players who played 90 minutes and a player who played 60 minutes I would expect to see lower 'per minute' outputs for those who played 90 mins due to the longer duration and fatigue in the later stages. Are you comparing the physical outputs for the first 60 minutes of the 90 mins for the players who played the full game, to the 60 mins outputs for those player who only played 60 mins? I don't think you will see a difference between substituted players and players who played the full game if you do the analysis this way.

Methods:

Why have you chose to compare the 1st three rounds of fixtures after the break with the 8th, 9th, 10th rounds at the start of the season? Would it not make sense to compare the 1st three rounds at the start of the season with the 1st 3 back after the break? 

Line 110 - the references 28 and 29 are not references in the text properly.

Section 2.3 - more clarity and detail is required around data collection here. I should be able to repeat the methods from your explanations, but I don't feel I could if I wanted to.  

Results: 

This section needs to be cut down for me. This should be a more concise section. The tables are good but the text part has too many words. 

Discussion:

This section needs significant review of the english used throughout. Sections do not make sense, although I think I can see what you are trying to say. I think the content is good (if it is what I think it is that you're trying to say) but the english needs revising. 

Conclusion:

Again, I would remove the 'spectacle' parts and focus on actual physical outputs and measurable variables that have/ have not changed. 

Author Response

First of all, we would like to thank you for your decision on the manuscript and for all the considerations and comments made to improve the quality of our article. The reviewer's comments are marked in black, the authors' response in blue and the changes made to the paper in red.

Abstract: 

I feel the abstract needs to be re-written. This section doesn't summaries the article well enough and is generally poor. I think you need to remove the mention of the 'spectacle'. This is a scientific article. This should be focused on changes in quantifiable measures like the physical or technical outputs of players, not a subjective assessment. The spectacle of the game could be very different simply because of the team which a person supports

We agree with the suggestions made, therefore, we have removed the colloquial word spectacle and explained in a better way the power of the study in terms of the analysis of the physical performance caused by the new rule of the increase in the number of changes in the comparison of the pre and post covid periods.

Introduction: 

In general, this section doesn't read well despite some good parts to it. The english needs significant revision to make this section read better. 

The English language has been reviewed and sent to a professional proofreader for analysis and modification.

I'm not sure about the velocity thresholds that you have used for high intensity distance, very high intensity, and sprint distance. This is different to those universally used in soccer and utilised in most research in these areas. 

We understand the concern and have proceeded to explain it better. The justification for choosing these ranges is that they are the ranges used by Mediacoach, the company that supplies LaLiga's data and which are delivered to the different clubs. It should also be noted that the metrics vary and are currently adapted by the teams' coaches according to their interests.  We have tried to combine them in a generic way using the data provided by LaLiga.

Changes made in line 83.

The activities performed by the players were classified into categories according to the following speed thresholds [13] (I) standing (0-0.6 km-h-1), (II) walking (0.7-7.1 km-h-1), (III) jogging (7.2-14.3 km-h-1), (IV) running (14.4-19.7 km-h-1), (V) high-intensity running (19.8-25.1 km-h-1), and (VI) sprinting (>25.1 km-h-1). High-intensity running consisted of running, high-speed running and sprinting (running speed >14.4 km-h-1), while very high-intensity running represented the sum of high-speed running and sprinting (running speed >14.4 km-h-1), while very high-intensity running represented the sum of high-speed running speed and sprinting (running speed >19.8 km-h-1). Maximum running speed was defined as the maximum speed a player reached during the match.

Lines 75-78 - I think you need to look in to this comment further. If you are looking at the difference between players who played 90 minutes and a player who played 60 minutes I would expect to see lower 'per minute' outputs for those who played 90 mins due to the longer duration and fatigue in the later stages. Are you comparing the physical outputs for the first 60 minutes of the 90 mins for the players who played the full game, to the 60 mins outputs for those player who only played 60 mins? I don't think you will see a difference between substituted players and players who played the full game if you do the analysis this way.

We appreciate the comment made by the reviewer. It has been amended to make the wording clearer. In reference to the second part of the comment, the authors believe that there is a difference between those who play 60 and those who play 90. Because the players who go out on the field have better values than those who play 60 and those who play 90 and if the sum is added up, that is to say taking the relative value of the 60 plus the 30 with the 90, they also have better values.

Moreover, the physical performance of players who enter the field of play replacing a teammate has been studied by different authors [24,29–31]. In general, and under normal game conditions, substitutes have shown better data in high-intensity trips than those players they replace and those who play the entire game [30–33]. The position within the team structure [24,29–31] the result of the match [30] and the time of the season [31] can also influence the performance of substitutes. In addition, it has been shown that substitute players have a lower workload and volume, considering both factors, the games in periods of congestion and the training sessions themselves in these periods in which the intensity of daily work decreases [34]. This shows that substitutions increase the possibility of maintaining and even improving the physical performances of the players.

Methods:

Why have you chose to compare the 1st three rounds of fixtures after the break with the 8th, 9th, 10th rounds at the start of the season? Would it not make sense to compare the 1st three rounds at the start of the season with the 1st 3 back after the break? 

Line 110 - the references 28 and 29 are not references in the text properly.

Section 2.3 - more clarity and detail is required around data collection here. I should be able to repeat the methods from your explanations, but I don't feel I could if I wanted to.  

The justification is present within this section, although we know that the methodology aspect was not clear enough. The only 3 matchdays that met this requirement before the modification were taken (the first three matchdays of the season could not be analysed as they did not meet this premise). Subsequently, we analysed the first 3 matchdays where this modification of substitutions did occur.

Taking into account the rest of your considerations, the structure of references 28 and 29 has been modified, and a figure explaining data collection and analysis has been added, to make it easier to identify and perform analysis with the same procedure and exclusion criteria.

Results: 

This section needs to be cut down for me. This should be a more concise section. The tables are good but the text part has too many words. 

Among the comments made to the results section, we welcome comments on the tables. We agree that it is a long section to interpret with respect to the text, but following the traditional writing methodology for this section, we must report all the results in writing, although we believe that the tables faithfully represent the summary of this section.

Discussion:

This section needs significant review of the english used throughout. Sections do not make sense, although I think I can see what you are trying to say. I think the content is good (if it is what I think it is that you're trying to say) but the english needs revising. 

We are grateful for your comment and therefore the English has been rewritten by professional reviewers.

Conclusion:

Again, I would remove the 'spectacle' parts and focus on actual physical outputs and measurable variables that have/ have not changed. 

It has been amended and corrected, removing the word spectacle and placing greater emphasis on quantifiable data.

Reviewer 3 Report

First of all, I would like to congratulate the authors for their study. The study is technically well arranged and written. However, the fact that the 5 substitutions rule has already been implemented by FIFA and this stuaiton reduces the originality of the study. By the way, substitutions are not regulated solely according to competition running mechanics. In addition, the game should not stop too much and the flow of the game should not be affected too much. Moreover, a separate study is required to determine the effect of substitutions on the players injuries. These issues need to be explained in the study. The title should be rearranged because of only running mechanics performance was measured in the study

Author Response

First of all, we would like to thank you for your decision on the manuscript and for all the considerations and comments made to improve the quality of our article.

We welcome your comments on the study. Taking into account your considerations, we have modified the title and some sections to specify that it is about physical performance.

We agree with you that the topic is relevant, and we believe that the selection of congested fixtures (higher number of matches in a short period of time) can provide an insight that will help coaches, coaching staff and regulatory institutions to have more information to argue their position.

We take note of their ideas to analyse them in future studies, we believe they can be interesting to provide more information on this type of periods.

Thank you for your consideration

Round 2

Reviewer 1 Report

Dear authors,

Congratulations on the improvement of the manuscript.

Based on the changes, I am in favor of accepting it.

Author Response

Thank you very much for your time in proofreading the paper. Thanks to your contributions we consider it a better job.

Reviewer 2 Report

'football' should be replaced for 'soccer' though out the document as 

Line 49 - remove 'therefore'. Start the paragraph with 'In recent years'

Line 55 - remove '...'

Line 89-93 - please re-read and amend. I think there is duplication in here. 

Line 106 - remove 'moreover'

Line 107 - remove 'and' after 'in general,' 

Line 140-145 - i think this needs re-reading, I don't think these sentences make sense. 

Line 147 - can you add in a line here to say that you were restricted to these speed thresholds by the system used in the league. This would help readers understand why these thresholds are different to the majority of the research in this area

Discussion/ conclusions - this section still needs revision with regards to the english language used. I think the points you try to make are ok but it is not written well enough for publication in my opinion. In your reply to the last review comments you said that these sections have been rewritten by 'professional reviewers', I don't think these people have helped improve the article. 

As I said last time, I really like the idea of the article and the question you are trying to address but the quality of the text is not good enough at the moment unfortunately. 

Author Response

All issues have been addressed and resolved. We apologise that in the version sent, due to the time limit for uploading the work, the work of the English proofreaders had not yet arrived. I now believe that all the language errors have been corrected.

It has been decided not to change the term because the same English-speaking reviewers have put it that way and because it is a term more commonly used in the English-speaking world, the ones who created football.

Line 140-145 - i think this needs re-reading, I don't think these sentences make sense. 

They are part of the exlusion criteria, they are marked as such to give evidence of the n that we discard; they are also added (without n) in the figure. This way you can see the number of substitutions before the new rule (i.e., 0, 1, 2 or 3 substitutions that could be made) and those after the new rule (i.e., 0, 1, 2, 3, 4 and 5 substitutions that could be made).

Thank you very much for your time. We believe that with your contributions the paper has improved ostensibly.